# SARS-CoV-2 Infection, Sex-Related Differences, and a Possible Personalized Treatment Approach with Valproic Acid: A Review

**DOI:** 10.3390/biomedicines10050962

**Published:** 2022-04-21

**Authors:** Donatas Stakišaitis, Linas Kapočius, Angelija Valančiūtė, Ingrida Balnytė, Tomas Tamošuitis, Arūnas Vaitkevičius, Kęstutis Sužiedėlis, Daiva Urbonienė, Vacis Tatarūnas, Evelina Kilimaitė, Dovydas Gečys, Vaiva Lesauskaitė

**Affiliations:** 1Laboratory of Molecular Oncology, National Cancer Institute, 08660 Vilnius, Lithuania; kestutis.suziedelis@nvi.lt; 2Department of Histology and Embryology, Medical Academy, Lithuanian University of Health Sciences, 44307 Kaunas, Lithuania; linas.kapocius@lsmuni.lt (L.K.); angelija.valanciute@lsmuni.lt (A.V.); ingrida.balnyte@lsmuni.lt (I.B.); evelina.kilimaite@lsmuni.lt (E.K.); 3Department of Intensive Care Medicine, Lithuanian University of Health Sciences, 50161 Kaunas, Lithuania; tomas.tamosuitis@lsmuni.lt; 4Institute of Clinical Medicine, Faculty of Medicine, Vilnius University Hospital Santaros Klinikos, Vilnius University, 08661 Vilnius, Lithuania; arunas.vaitkevicius@santa.lt; 5Department of Laboratory Medicine, Medical Academy, Lithuanian University of Health Sciences, Eiveniu 2, 50161 Kaunas, Lithuania; daiva.urboniene@lsmuni.lt; 6Institute of Cardiology, Laboratory of Molecular Cardiology, Lithuanian University of Health Sciences, Sukileliu Ave., 50161 Kaunas, Lithuania; vacis.tatarunas@lsmuni.lt (V.T.); dovydas.gecys@lsmuni.lt (D.G.)

**Keywords:** valproic acid, COVID-19, sex differences, pre-clinical research, clinical research

## Abstract

Sex differences identified in the COVID-19 pandemic are necessary to study. It is essential to investigate the efficacy of the drugs in clinical trials for the treatment of COVID-19, and to analyse the sex-related beneficial and adverse effects. The histone deacetylase inhibitor valproic acid (VPA) is a potential drug that could be adapted to prevent the progression and complications of SARS-CoV-2 infection. VPA has a history of research in the treatment of various viral infections. This article reviews the preclinical data, showing that the pharmacological impact of VPA may apply to COVID-19 pathogenetic mechanisms. VPA inhibits SARS-CoV-2 virus entry, suppresses the pro-inflammatory immune cell and cytokine response to infection, and reduces inflammatory tissue and organ damage by mechanisms that may appear to be sex-related. The antithrombotic, antiplatelet, anti-inflammatory, immunomodulatory, glucose- and testosterone-lowering in blood serum effects of VPA suggest that the drug could be promising for therapy of COVID-19. Sex-related differences in the efficacy of VPA treatment may be significant in developing a personalised treatment strategy for COVID-19.

## 1. Introduction

The β coronavirus pandemic, named severe acute respiratory syndrome coronavirus 2 infection (COVID-19), has led to calls to identify effective drugs to treat the disease [1]. The COVID-19 strategy for treating severe diseases is inextricably linked to the development of registered medicines for this new therapeutic indication [2,3].

Most COVID-19 patients have a mild to moderate condition, while some have progressed to a critical condition. SARS-CoV-2 virus mainly affects the lungs, causing respiratory failure and secondary hypoxemia in one-fifth of hospitalised patients [4,5]. Severe illnesses may be associated with cardiovascular complications, thrombus formation, septic shock or acute kidney injury [6,7,8]. SARS-CoV-2 exacerbates disease progression and organ damage due to an uncontrolled immune response in a “cytokine storm” [9,10,11]. Autopsy studies have shown that SARS-CoV-2 RNA has been detected in cells from the lung, trachea, kidney, liver, brain, intestine, testis and blood tissues, indicating SARS-CoV-2 multiorgantropism [12,13,14]. Pathological changes were in arterioles/venules, capillaries and medium-sized blood vessels of the affected organs, mainly due to the accumulation of lymphocytes, plasma cells and macrophages around the endothelial cells: endotheliitis leads to cell apoptosis, tissue edema, thrombotic microcirculatory pathology, vasoconstriction and ischemia [7,14].

The meta-analysis of published global cases (analysed 107 reports from around the world) shows that men and women are at equal risk of SARS-CoV-2 infection. Men have a higher risk of severe COVID-19 are three times more likely to be treated in an intensive care unit [15]. Male mortality is elevated in all age groups and most pronounced in middle age [16,17]. A study of 17 million adults confirms a significant association between male sex and the risk of death from COVID-19 [18,19,20,21].

The median viral RNA content of nasopharyngeal swabs and saliva was higher in men than women [22]. Sex-specific immune reactions are thought to determine the progress of COVID-19 [23]. Sex-driven differences are based on a women’s more effective early adaptive immune response [24,25,26,27]. Sex-related disparities in disease progression may be due to estrogen-induced reduced expression of the angiotensin-converting enzyme 2 (ACE2) receptor [28,29], which serves as a gateway for SARS-CoV-2 to enter the target cell [30,31,32].

The novel bioinformatic approach includes a wider range of clinically approved drugs so that more possibilities are allowed for them to repurpose against COVID-19 [33]. One such drug is valproic acid (2-n-propyl-pentanoic acid; VPA) [34]. VPA is a histone deacetylases (HDACs) inhibitor [35]. VPA is a commonly prescribed antiepileptic drug [36], which is also used to treat bipolar disorder [37], schizophrenia and various forms of headache [38,39,40]; it is an investigational anti-cancer preparation as an immune modulator [33,41]. VPA has an extensive research history in treating various viral infections [33,42]. VPA has been used in clinical practice for six decades, and has a well-known safety profile, as well as therapeutic serum concentrations that make it an attractive drug for adjunctive therapy in off-label settings.

Research of sex-specific features may lead to a new approach to the COVID-19 treatment. This review aims to evaluate VPA as a potential medicine for the treatment of COVID-19 and to elucidate the possible biological sex-related mechanisms of pharmacology, to review VPA as a potential drug to prevent the progression of COVID-19 and to provide personalised treatment of the disease.

## 2. VPA Metabolism and Sex

VPA is completely absorbed; bioavailability is ≥80% [43]. VPA molecules are 87–95% bound to plasma proteins, resulting in low drug clearance [44]. The free form of VPA crosses the cell membrane [43]. VPA peak time in plasma is 4 h; the half-life is 11–20 h, and depends on the drug formulation [45]. With continuous oral administration, the plasma concentration of VPA is 280–700 μmol/l [43]. In adult humans, the main metabolic pathways of VPA are 50% glucuronidation, 40% mitochondrial β-oxidation and a small fraction by cytochrome P450-mediated oxidation [44]. Urinary excretion of intact VPA is <3% [43].

The G protein system (MRP) is involved in the intracellular transport of VPA; drug transport via G protein substrates is higher in females than in males of experimental animals and humans [46,47], as testosterone down-regulates the drug transport [48]. Hepatobiliary transport of VPA, bypassing hepatic metabolism and subsequent reabsorption from the duodenum after biliary excretion, is stronger in females. The proportion of the bioavailable dose of VPA reabsorbed was 2.1-fold lower in males than in females, indicating a significant difference in hepatobiliary drug transport [49]. In males, due to lower expression of efflux pumps, a saturation of hepatobiliary transport results in longer retention of VPA in the hepatocyte, leading to expantion in VPA clearance [50]. In women using hormonal contraception, the pharmacokinetic parameters of VPA were similar to those in men [49].

## 3. SARS-CoV-2 Virus and VPA

The docking, binding energy calculation determines that VPA metabolite 4-ene-VPA-CoA creates a stable interaction with nsP12 of SARS-CoV2 RNA polymerase and VPA-CoA could specifically inhibit the target. SARS-CoV-2 RNA polymerase is an enzyme playing in viral RNA replication and the virus’s survival in a host [51,52]. The SARS-CoV-2 virus x-ray crystal structure of a critical protein in the virus’s life cycle is the central protease (M^pro^, 3CL^pro^) [53,54,55]. The M^pro^ importance recognises M^pro^ as a target for antiviral drugs, designed as a virus 3CL^pro^ inhibitor, for COVID-19 therapy [56,57]. HDACs’ inhibitors are tightly bound into the active site of the crystallographic virus M^pro^ structure [58]. The SARS-CoV-2 protease NSP5 interacts with the HDAC2. Researchers predict that NSP5 may inhibit the transportation of HDAC2 into a nucleus, and could affect the HDAC2 strength to interfere with the interferon response and inflammation [59,60]. Experimental studies show that the binding of HDAC2 to the promoters was lower in females than in males [61]. The HDAC2 activity can not only be modulated by VPA binding to the catalytic center, but the HDAC2 protein level is susceptible to selective regulation by VPA [62]. VPA blocks the zinc-containing catalytic domain of HDACs [63]. VPA treatment reduced HDAC2 level in male rats’ brain frontal cortex tissue, but no VPA effect was for HDAC2 protein in females [64].

## 4. Sex-Related AEC2 Expression and VPA Effect

The SARS-CoV-2 virus connects to the cell membrane-bound ACE2, a functional receptor for SARS-CoV-2, to mediate virus entry into human cells [65,66]. The binding capacity of the virus S protein to the ACE2 receptor was 10–20-fold higher than that of other SARS-CoV [67]. ACE2 is a single-pass type I membrane protein with an active domain exposed on the cell surface in the lung, kidney, arteries, heart and intestine tissue [68,69,70]. The ACE2 gene is encoded in the X chromosome [34,71]; this may counteract X-inactivation in women [72]. Male cells always hold an X chromosome, and display a single ACE2 allele. The female X chromosomes mosaicism is related to the heterogenic ACE2 allele given among cells. A potentially more efficient form of ACE2 receptor would have half all cells in females; therefore, some alleles of this gene may code for the receptors with different efficiency of binding SARS-CoV-2 virus, providing them a partial resistance to the COVID-19 infection [73]. A lower ACE2 tissue expression was observed in women than men [32,74,75]. ACE2 activity is lower in older women than in young ones, while the same does not happen in males [75,76]. ACE2 protein expression in the lung was higher in adult non-smoking males than women [32]. In male smokers, it was more increased than in female smokers, and there was a notably elevated ACE2 level in nasal and bronchial airways cells of male smokers [77,78,79]. Testosterone increases ACE2 levels, whereas estrogens maintain suppressed expression of ACE2 [29,80,81]. Pharmacologic intervention with an androgen receptor antagonist significantly suppressed ACE2 expression in male mice’s lungs [77]. VPA down-regulates ACE2 gene expression [82]. The treatment of human umbilical vein and human coronary artery endothelial cells with VPA in vitro significantly reduced ACE2 expression [82].

## 5. Virus Entry into Cell and VPA

Co-expression of ACE2 and the transmembrane serine protease 2 (TMPRSS2) receptor is required for SARS-CoV-2 infection of cells [68]. Human lung tissue, type I and II alveolar epithelial cells, arterial and venous endothelial cells, cardiomyocytes, arterial smooth muscle cells expressing ACE2 are targets for the SARS-CoV-2 virus [70]. VPA can reduce ACE2 in endothelial cells [77,83]. TMPRSS2 is highly expressed in lung tissue cells, and makes the respiratory system susceptible to the virus [84]. The cell-surface TMPRSS2 is operated for virus-S protein priming and the activation of membrane fusion processes for the SARS-CoV-2 [68,85]. TMPRSS2 is potentially the most promising target for COVID-19 therapy, as its specific expression in the alveolar cells [86]. VPA reduces the expression of TMPRSS2 [86]. TMPRSS2 is upregulated by androgens [87], and could appear linked to the increased risk of COVID-19 infection in men [8]. There was no sex-related difference in *TMPRSS2* expression in humans or mice’s lungs [77]. TMPRSS2 inhibitors are effective against the SARS-CoV-2 [68]. VPA reduced TMPRSS2 expression in prostate cancer cells [88]. After the virus is connected to ACE2, the ACE2 extracellular domain controlling the catalytic effect is cleaved by ADAM-17 (a disintegrin and metalloproteinase-17), enabling viral transport to the cytoplasm [89]. TMPRSS2 competes with the ADAM17 for ACE2 processing [85]. The enhanced activity of ADAM17 in males was reported [90]. ADAM-17 cleaves ACE2 bearing membrane and releases soluble ACE2 (sACE2) into the circulation [91]. The sACE2 retains activity, and can partially block SARS-CoV-2 binding to ACE2 of the target cell membrane; sACE2 could reduce viral replication [92]. Cigarette smoking causes decreased sACE2 blood levels [93].

## 6. Sex-Related COVID-19 Infection Progression Mechanisms and VPA

Due to the high viral infection load, membrane ACE2 and its mRNA expression are significantly diminished in COVID-19 patients [69,94,95,96]. At a later COVID-19 infection stage, down-regulated ACE2 in tissues may worsen the imbalance in the renin-angiotensin system (RAS). ACE2 has a protective effect through RAS regulation [97], and protects against RAS-mediated activations of harmful effects [69,98]. Depleting ACE2 in tissue cells leads to an increase in angiotensin II (Ang II) blood serum level and the activation of the AT_1_ receptors, which would activate ADAM17 more. The ACE2 expression is transcriptionally suppressed due to AT_1_ activation [99]. Increased Ang II levels act as a vasoconstrictor and a pro-inflammatory molecule through AT_1_ [100]. ACE2 knockout results in pathology similar to the acute respiratory distress syndrome in mice [101]. The ACE2 molecule reduces RAS activity by converting Ang II into Ang 1–7 [102,103,104], decreasing Ang II level and the AT1 activation, which manages reduced pathological inflammation effects in tissues [69,105,106]. Sex distinctions of the RAS in response to stimulation and inhibition of the system have been reviewed [81,107,108]. Higher levels of ANG (1–7) in women may inhibit the harmful effects of ANG II and its activation [109]. Compared with female rats, males have higher AT1 receptor RNA, higher AT1 protein levels, higher receptor density in kidneys and ∼40% higher specific AT1 binding in the glomeruli than females. These differences are 17β-estradiol (E2) dependent [81,108,110]. Activation of AT1 mediates ANG II’s biological functions, such as sodium reabsorption, vasoconstriction, increased oxidative stress and inflammation [111,112]. VPA, inhibiting HDAC1 and HDAC2, down-regulates Ang II and AT1 activity [113,114]. VPA reverses the ANG II-induced increment of HDAC2 RNA and protein levels in cardiomyocytes [115]. Anti-hypertensive action of VPA is mediated by the inhibition of HDAC1 via acetylation processes [113,116].

## 7. Pre-Clinical Research of VPA Efficiency on Inflammation Mechanisms

### 7.1. VPA Effectiveness by Experimental Models In Vivo

As a multifunctional regulator of innate and adaptive immune cells, VPA reduces macrophage infiltration in various models of inflammation (Table 1). VPA attenuated the significant upregulation of profibrotic and proinflammatory genes, the deposition of collagen and the infiltration of macrophages into the kidney [117]. VPA significantly reduced cigarette-smoke-induced neutrophil influx in the female Balb/c mice lung inflammation model, working as the endopeptidase (PE) inhibitor and potentially serving as therapeutic in inflammatory lung disorders, causing a decline in neutrophil infiltration in the bronchoalveolar fluid in a chronic lung inflammation model in female A/J and Balb/c mice [118]. VPA impaired M1 macrophage proliferation while promoting the accumulation of M2 macrophages in the Wistar male rats lung injury model with the lessened inflammation and enhanced tissue repair [119]. VPA can attenuate acute MAP kinase activation in the lung in a sublethal model of hemorrhagic shock reduces pulmonary neutrophil infiltration 20 h after blood loss [120]. In female BALB/c mice LPS-activated dendritic cells (DCs), VPA caused the decreased TNF-*α*, IL-1*α*, IL-1*β*, IL-1RA, IL-6, IL-7, IL-10, IL-12p40, IL-12p70, IFN-*γ* and TGF-*α* production. VPA was lowering immune cells’ capacity to induce a pro-inflammatory response may offer new therapeutic options for managing septic shock [121]. VPA attenuated the clinical severity of Coxsackie B3 virus (CVB3) myocarditis, and mortality from CVB3-induced myocarditis of BALB/c male mice, decreased the percentage of splenic Th17, increased the rate of Treg cells, downregulated the IL-17A expression, upregulated IL-10 in serum and heart tissues of CVB3 infected mice, inhibited the differentiation of the Th17 cells and promoted the differentiation and suppressive function of Treg cells [122]. In the mouse NIH3T3/BL6 post-operative inflammation model of conjunctival scarring, VPA repressed the CD45^high^F4/80^low^ macrophage subset, repressed the CXCL1, IL-5, IL-6 and IL-10, tissue NF-кB2 p100 protein generation in males and females [123]. VPA decreases macrophages infiltration, apoptotic cell death and caspase 3 activation, reduces the lesion volume, improves active recovery after rat male spinal cord injury, improves functional recovery by attenuating the blood-spinal cord barrier after spinal cord injury by inhibition of MMP-9 activity [124]. In the acute colitis model, disease amelioration by VPA was associated with prevention from weight loss, a decrease in histological signs of inflammation, suppression of the pro-inflammatory cytokines IFN-γ and IL-6 [125]. In the female C57BL/6 mouse model, VPA diminished CD4+ T-lymphocyte infiltrates, associating with caspase 3-mediated apoptosis [126]. VPA reduces pro-inflammatory cytokines and reactive oxygen species (ROS) levels and attenuates rat’s multiple organ damages induced by LPS-provoked septic shock due to the recovery of histone H3 acetylation, including: attenuated alanine aminotransferase, aspartate aminotransferase, urine nitrogen, creatinine serum level and decreased TNF-α and myeloperoxidase levels in lung tissue of male rats [127]. VPA treatment improves early survival, lung, liver and brain function in highly lethal poly-trauma and male rat hemorrhagic shock models [127,128]. VPA inhibited by 92% leukocytes migration to the peritoneal cavity in a male rat peritonitis-induced model by decreasing TNF-α and IL-1β, IL-6 levels and ROS generation, and the effect was similar to indomethacin force [129]. In the ischemic kidney/reperfusion injury rat model, VPA-treated male rats show a significant increase in blood IL-10 and TGF-β mRNA levels, a direct correlation between IL-1β and TNF-α mRNA expression and IL-10 with TGF-β levels indicating the anti-inflammatory VPA effect, histopathology showed decreased kidney ischemic changes and the reduced serum creatinine level in VPA-treated animals [130].

### 7.2. VPA Effectiveness by Experimental Studies In Vitro 

VPA drastically inhibited the multiplication of the enveloped viruses (zoonotic lymphocytic choriomeningitis, West Nile viruses). While it did not affect infection by the non-enveloped viruses, VPA abolished West Nile RNA and protein synthesis, indicating that VPA can interfere with the viral cycle at different steps of enveloped virus infection. VPA reduced vesicular stomatitis virus infection [135]. VPA reduces the production of TNF-*α*, IL-6 in female BALB/c mice bone marrow-derived macrophages [121]. VPA polarises mouse macrophage cell line RAW264.7 and primary mouse bone marrow macrophages (BMMs) of C57BL/6 and BALB/c female mice from a pro-inflammatory M1 to an anti-inflammatory M2 phenotype [137]. VPA inhibited macrophage-mediated T helper 1 (Th1) effector, enhanced Th2 effector cell activation affects the macrophage function, repressed the production of IL-12 and TNF-α by LPS-induced macrophage activation, and promoted IL-10 expression. VPA also affected the costimulatory molecule expression on LPS-stimulated RAW264.7 and BMMs; it downregulated CD40 and CD80 and upregulated CD86 [137]. VPA effect on LPS-induced inflammatory response in mouse RAW 264.7 macrophage-like cells shows that VPA down-regulates LPS-induced NF-κB-dependent transcriptional activity via impaired PI3K/Akt/MDM2 activation and enhanced p53 expression [140]. The VPA-treated fibroblasts resulted in diminished CCL2, VEGF-A, IL-15 levels and in the presence of TNF-α, VPA inhibited the induction of CCL5 and VEGF-A and NF-кB2 p100 level [123]. VPA in human monocyte-derived macrophages (MDMs) infected with Dengue virus (DENV) reduced secretion of IL-8, IL-1b, IL-6, TNF-α and IL-10. Researchers suggest VPA modulates the cytokine storm associated with DENV disease and prevents progression to severe illness [134]. VPA inhibits NF-kappaB activation induced by LPS, inhibited LPS-induced production of TNF-alpha and IL-6 by human monocytic leukemia THP-1 cells [141]. VPA repressed the healthy volunteers’ monocyte differentiation into DCs, and disturbed the DCs differentiation and maturation [142]. VPA-treated healthy blood donors T CD8+ lymphocytes show a decrease in cellular proliferation [139].

## 8. VPA and Mitochondrial Function of Immune Cells

Mitochondria functions range from supplying energy activation of anti-viral and anti-inflammatory mechanisms [143]. The mitochondria may be affected by the SARS-CoV-2 main protease NSP5. NSP5 interacts with tRNA methyltransferase 1 (TRMT1); its gene is responsible for transferring a methyl group onto a guanine residue in mitochondrial tRNAs [60,144]. The TRMT1 cleaved by NSP5 leads to removing the TRMT1 zinc finger. TRMT1 knockout leads to increased sensitivity to oxidative stress [3,60]. Cells with decreased TRMT1 activity exhibit increased endogenous ROS production [60,144]. In male and female tissues, there was no significant difference in the expression of RNA TRMT1 [145]. VPA decreases oxidative stress by enhancing the enzymatic antioxidant system [146]. 

VPA metabolites significantly decrease pyruvate-driven oxidative phosphorylation in mitochondria by conflict with pyruvate transport, thus settling mitochondrial energy production [147]. VPA treatment significantly reduced SLC5A8 gene expression in gonad-intact and castrated male rat thymocytes, while in gonad-intact female rat thymocytes, VPA significantly increased gene expression. Higher SLC5A8 gene expression was found in thymocytes of gonad-intact male rats than in corresponding female rats [133]. SLC5A8 has a physiological function transporting short-chain fatty acids into the cell and regulates mitochondrial metabolism [148,149]. SLC5A8 activity has immunomodulatory effects by blocking the development of dendritic cells, making SLC5A8 essential for immune homeostasis through the release of cytokines [150,151]. SLC5A8 induces cell apoptosis by inhibiting pyruvate-dependent HDACs [152]. Inhibition of the SLC5A8 gene is associated with DNA methylation, and treatment with DNA demethylating agents increases SLC5A8 gene expression [153]. VPA can activate SLC5A8 genes regulated by DNA methylation [154,155]. VPA through mitochondria affecting immune cell metabolism, immune-related functions could polarise the pro-inflammatory M1 phenotype of macrophage to the anti-inflammatory M2, which is unable to induce naïve T CD4+ differentiation into a Th1 profile, favouring a Th2 phenotype by changing the kind of respiration from the down TCA cycle to β-oxidation [137,156,157]. Further research needs to examine how VPA-induced sex-specific changes in SLC5A8 capacity influence immune cell function.

## 9. Sex-Related Differences of Immune Response and COVID-19 

Gonad hormones affect the immunological response, with the estrogens being both pro-inflammatory and anti-inflammatory [158,159]. Testosterone has a suppressive impact on immune function [160]. The Y and X chromosomes modulate the immunity in COVID-19 infection [73]. The differentiation and maturation of innate immune cells, like neutrophils, macrophages, dendritic cells, natural killer cells and T cells, are modulated by gonad hormones [161]. The number of innate immune cells, including monocytes and macrophages, is higher in females [162,163]. Males have lower CD4+ and CD3+ cell counts, CD4+/CD8+ cell ratio than females [164,165,166,167]. Females exhibit higher cytotoxic T cell activity and increased expression of antiviral genes [25]. The immune cells Toll-like receptor 7 (TLR7) detects single-stranded RNA viruses. The TLR7 gene is encoded in the X chromosome. Higher expression of TLR7 level in females may serve to escape antiviral genes inactivation [168,169,170]. The 17β-estradiol (E2) regulates various physiological and pathophysiological changes in DNA by epigenetic mechanisms; E2 and VPA mediate immune responses and autoimmunity in women [171,172]. Sex-related differences in the immune response have been reviewed extensively [22,173]. In males over females, innate inflammatory cytokines and chemokines increase throughout the COVID-19 course [22]. COVID-19 severe patients exhibit substantially elevated plasma levels of pro-inflammatory cytokines [174,175], and activation of the release of cytokines may lead to cytokine storm [174,176] and multiple organ failure [177]. Thrombosis is indistinguishable from acute respiratory distress syndrome (ARDS) with micro-and macro-thrombosis [178]. VPA mitigates the inflammation and prevents acute respiratory distress syndrome in a murine model of Escherichia coli pneumonia [131].

## 10. COVID-19 Thrombotic Complications and VPA

### 10.1. SARS-CoV-2 and Sex-Related Thrombotic Complications

The pathophysiology of COVID-19 complications is characterised by clinical features of thrombosis and disseminated intravascular coagulopathy in the airways, myocardium, kidneys, brain and other organs [179]. Thrombosis is found in approximately 30% of COVID-19 hospitalised patients [180]. The incidence of thrombosis in COVID-19 is higher in men than in women and explains the higher mortality in men [181]. In an analysis of 29 studies, 70% of all thromboembolic events occurred in men and 30% in women [182]. Viral invasion due to severe vascular endothelial damage triggers the coagulation cascade, impairs fibrinolytic activity, releases von-Willebrand factor [13], increases total cytokine release, activates platelets and the complement system and generates thromboxane [183,184]. SARS-CoV-2 can directly activate coagulation via the viral M^pro^; the active site of M^pro^ is structurally similar to the active site of FXa and thrombin and can therefore activate coagulation [185]. The development of thrombosis has been attributed to the direct effects of the virus by increasing the levels of pro-inflammatory cytokines and pro-inflammatory M1 macrophages, by activation of the complement system and by endothelial dysfunction, leading to disseminated intravascular coagulopathy [186,187,188,189]. Endothelial dysfunction and its association with thrombosis have been implicated in SARS-CoV-2-induced target organ damage [190]. VPA binding to SARS-CoV-2 M^pro^ is expected to inhibit M^pro^ pathways [191]. Older men with hypertension, chronic kidney disease, coronary disease, diabetes mellitus and obesity are at increased risk of thrombotic complications [192,193,194]. Changes in plasma levels of D-dimer, von Willibrand factor (vWF), fibrinogen, tissue-type plasminogen activator (t-PA), plasminogen activator inhibitor-1 antigen (PAI-1) antigen are associated with poorer outcomes in COVID-19 patients [195,196,197,198].

### 10.2. VPA Effect on Thrombosis Mechanisms, COVID-19

The VPA effects on thrombogenesis have been explored in pre-clinical studies and during the treatment of patients with VPA (Table 2). HDAC inhibitors have reduced platelet counts and inhibit platelet function [199,200], while other VPA experimental and clinical trials did not find such effects [201,202]. The baseline platelet count was similar in women and men. A causal relationship between prolonged use and rising plasma VPA levels and reduced platelet counts, with reversal of thrombocytopenia after reduction of VPA dosage, was reported: that of thrombocytopenia substantially increased at VPA levels above 100 ug/mL in women and above 130 ug/mL in men; women were significantly more likely to develop thrombocytopenia [203]. There is a significantly higher female overrepresentation in heparin-induced thrombocytopenia, with females at approximately twice the risk of thrombocytopenia than males. However, the underlying mechanism for this sex difference is unclear [204]. VPA may affect several different coagulation factors: decrease in von Willebrand factor:antigen (vWF:Ag) concentration [205,206]; protein C level [205,207], protein S level [207,208], antithrombin III level, decrease prothrombin time [205,206] and increase activated partial thromboplastin time [205,206,207].

SARS-CoV-2 activates the complement system, either directly or through an immune response. Activated complement promotes inflammation [212]. Complement activation is increased and constant in severely ill COVID-19 patients, and complement activation is via the alternative pathway (AP) [213]. The anaphylatoxins C3a and C5a are significant contributors to the cytokine storm syndrome [214]. The healthy adult population is characterised by substantial sex-related differences in complement levels and function: significantly lower AP activity was in females than males. AP revealed lower C3 levels in women [215]. In experimental intestinal ischemia with an acute inflammatory response, complement activity was sex-dependent: female MBL-/- and P-/- mice had significantly less C5a in their serum than males [216]. Experimental results indicate that lysine acetylation by VPA is associated with attenuated C3 gene expression. VPA-associated reductions in circulating complement and clotting factors result from changes in liver-specific gene expression [209]. VPA inhibits intercellular adhesion molecule-1 (ICAM-1) and E-selectin [83,211]. Analysis from patients hospitalised with COVID-19 showed higher circulating VCAM-1 and E-selectin levels in men than women [217]. The endothelial cell adhesion molecules elevated levels promote tissue infiltration of circulating leukocytes and are associated with inflammation and thrombosis, which occur at a higher frequency in males [218]. VPA reduced endothelial cell dysfunction through the mechanisms of action of transforming growth factor-β (TGF-β) and vascular endothelial growth factor (VEGF) in a porcine model of ischemia-reperfusion of hemorrhagic shock [219]. Inhibiting TGF-β activity, VPA alleviates pulmonary fibrosis through epithelial-mesenchymal transition inhibition in vitro and in vivo [132]. Estradiol has been shown to decrease TGF-β1 synthesis [220]. VPA inhibiting IL-12 and TNF-α, reversing macrophage polarisation from pro-inflammatory to the anti-inflammatory type, and reducing macrophage infiltration reduces the risk of thrombosis [117,137].

### 10.3. VPA and Fibrinolysis

Treatment with VPA in a rat thrombosis model reduced thrombus formation and did not increase bleeding tendency [201]. VPA can selectively manipulate the fibrinolytic system to reduce thrombus formation in blood vessels in vivo. In a murine model of thrombosis induced by intravascular injury, VPA treatment increased t-PA production in blood vessels [201,210,221,222,223] was associated with less fibrin accumulation and fewer thrombi [201,224]. Impaired fibrinolysis, due to reduced t-PA production and depleted storage or increased expression of a significant inhibitor of fibrinolysis PAI-1 [225,226], has been reported in coronary heart disease patients with cardiovascular risk factors, such as hypertension and obesity [227,228,229,230,231]. A clinical trial of VPA treatment showed a significant reduction in PAI-1 and signs of improvement in fibrinolysis, favourably altered the balance between t-PA and PAI-1, and the dose of VPA treatment was significantly lower than the usual dose of VPA for epilepsy [202,223,224]. Thus, VPA could be a potential alternative for preventing thrombotic events based on improved endogenous fibrinolysis [202].

## 11. Discussion

Sex-related differences in the COVID-19 progression and complications rate suggest that sex biological factors are important in the pathogenesis of COVID-19. Identifying the association of sex-specific factors with associated differences in risk of COVID-19 unfavourable outcome is essential for the development of effective personalised treatment. Detailed knowledge of the mechanisms underlying the differences in immune response between women and men, which may also be related to the risk of thrombotic complications, should lead to new therapeutic strategies.

In this review, we could not provide more detailed information on sex differences in the effects of VPA, as most of the studies involved animals only of one sex or even without specifying the sex of the animal or the cells. In some cases, patients or cells of different sex were combined without addressing sex differences. Regulatory guidelines for pharmaceutical research call for assessing the influence of sex on drug effectiveness, and state that the drug development should provide adequate information on the efficacy of drugs in relationship to sex [232,233]. Clayton points out that female and male cells differ in response to chemicals. Therefore, in pre-clinical and clinical trials, the sex-related effects of investigational drugs cannot be ignored [234,235].

Inflammation alters the ratio of histone acetyltransferases to HDACs and in-vitro or in-vivo data suggest that HDAC inhibitors may be anti-inflammatory agents [236]. VPA exerts organ protection from viruses through multiple anti-inflammatory pathways. Sex differences in the expression of genes related to mitochondrial metabolism may indicate a possible involvement of mitochondria in the susceptibility of infection to VPA treatment. Virus replication and survival depend on the energy produced by the cell’s mitochondria, so antiviral therapies may include drugs that alter the mitochondrial energy mechanisms [237]. The major SARS-CoV-2 protease NSP5 can affect mitochondria by interacting with HDAC2 [3]. VPA, inhibiting HDAC activity [238], causes a decrease in DNA methylation of the SLC5A8 gene [153]. Short-chain fatty acids transported into immune cells via SLC5A8 alter HDAC activity; SLC5A8, participating in the mitochondrial β-oxidation pathway, regulates mitochondrial metabolism [150,151]. VPA treatment had an opposite effect on SLC5A8 RNA gene expression in female and male rat thymocytes [133]. The French Ministry of Health has issued a pharmacovigilance alert that COVID-19 patients should not take Ibuprofen, as the drug may worsen the condition [239]. Ibuprofen is the most potent inhibitor of the SLC5A8 transporter; it is a specific blocker of SLC5A8 [240]. The European Medicines Agency (EMA) responded by saying: “EMA is monitoring the situation closely, and will review any new information that becomes available on this issue” [239]. Thus, the function of the SLC5A8 transporter may be of pharmacological relevance to the study of COVID-19 disease immune response in relation to sex.

Furthermore, pro-inflammatory-immune cells derive most of their energy from aerobic glycolysis to generate more energy and maintain increased activity [239,241]. Rapid growth and proliferation of virus-activated T cells require glucose uptake and glycolysis [242,243,244]. Elevated glucose level favours the progression of SARS-CoV-2 infection [245]. VPA treatment reduces blood glucose levels in animals and humans [246,247].

Data from experimental, epidemiological and clinical studies suggest that VPA has anti-platelet and anti-thrombotic effects. Clinical use of VPA in the treatment of epilepsy is associated with a lower risk of thrombosis, myocardial infarction and stroke [248,249,250,251]. Current anti-thrombotic therapies inhibit the coagulation cascade and platelet function, but dosing is not optimal to prevent bleeding complications [252]. Thus, VPA could be a potential alternative for preventing thrombotic events in COVID-19 patients based on improved endogenous fibrinolysis. The anti-thrombotic effect of VPA may be significantly related to its impact on suppression of the immune response, which may also be sex-related.

VPA treatment decreases serum testosterone levels [253], and in this respect, the relationship of VPA treatment to sex is an important area of research for COVID-19 treatment. Two studies of men undergoing hormonal therapy for prostate cancer show a potential protective effect of androgen suppression on the risk of severe COVID-19 [254,255]. ACE2 level in human alveolar epithelial cells can be downregulated by 5α-reductase inhibitors, suggesting an androgen-driven expression mode [256]. This link could pave the path to novel strategies, including re-purposing approved androgen synthesis inhibitors or androgen receptor antagonists to treat COVID-19. These strategies are the point of the NCT04374279, NCT04475601, NCT04509999 and NCT04397718 clinical trials [257,258]. The development of VPA on COVID-19 therapy is currently being investigated in clinical trials [259,260,261].

## 12. Conclusions

The anti-inflammatory, anti-thrombotic, immunomodulatory, serum glucose-lowering and testosterone-lowering effects of VPA suggest that it may be a promising investigational medicinal product for the treatment of COVID-19. The pharmacological mechanisms of VPA suggest that VPA could be a drug for the prevention of COVID-19 progression. 

The sex-specific differences in the course of COVID-19 and the mechanisms of action of VPA point to the need for prospective, controlled clinical trials to assess the sex-specific efficacy of valproic acid preparations.

## Figures and Tables

**Table 1 biomedicines-10-00962-t001:** Experimental studies of VPA treatment effectiveness on immune-inflammation in vivo and in vitro.

#	Experimental Model	Animals/Cells	Sex	VPA Treatment Effect	Ref.
1.	Lung injury model	Wistar rats	males	↓ M1 macrophage proliferation and ↑ the M2 macrophages proliferation in the rat lung; ↓ inflammation; ↑ tissue repair	[119]
2.	Lung inflammation model	Balb/c mice	females	↓ cigarette-smoke induced neutrophil influx; working as the PE inhibitor; ↓ inflammatory lung disorders	[118]
3.	Klebsiella pneumonia sepsis model	BALB/c mice	females	↓ immune cells capacity to induce a proinflammatory response	[121]
4.	Sublethal model of hemorrhagic shock	Wistar Kyoto rats	males	↓ hemorrhagic shock activated pro-inflammatory MAPK pathways; ↓ pulmonary neutrophil infiltration	[120]
5.	Chronic lung inflammation model	A/J mice	females	↓ neutrophils infiltration in the bronchoalveolar fluid	[118]
6.	Coxsackie B3 virus myocarditis model	BALB/c mice	males	↓ splenic Th17 and stimulated Treg cells; ↑ T cells apoptosis; ↓ IL-17A expression, ↑ IL-10 in serum and heart tissues; ↓ myocardial damage	[122]
7.	Post-operative inflammation the model of conjunctival scarring; Conjunctival inflammation model	NIH3T3/BL6 mice	males and females	↓ recruitment of a D45^high^F4/80^low^ macrophages; ↓ chemokine and cytokine levels in tissues; ↓ tissue NF-кB2 p100 levels; ↓ TNF-α induction of chemokines, cytokines and NF-кB2 p100 expression; ↓TNF-α stimulation of NF-кB	[123]
8.	Spinal cord injury model	BALB/c mice	males	↓ macrophages infiltration, apoptotic cell death and caspase 3 activation; ↓ the lesion volume and improved functional recovery after spinal cord injury	[124]
9.	Acute DSS-induced colitis model	C57BL/6J mice	females	disease amelioration was associated with prevention weight loss, ↓in histological signs of inflammation, ↓ IFN-γ and IL-6	[125]
10.	Experimental autoimmune encephalomyelitis model	C57BL/6 mice	females	↓ CD4+ T-lymphocyte infiltrates, associating with caspase 3 mediated apoptosis	[126]
11.	A model of LPS-provoked septic shock	Sprague-Dawley rats	males	↓ multiple organ damage caused by LPS induced septic shock	[127]
12.	Carrageenan-induced peritonitis model	Wistar rats	males	↓ by 92% leukocytes migration to the peritoneal cavity in a rat peritonitis; ↓ TNF-α, IL-1β, IL-6 levels, ROS generation	[129]
13.	Hemorrhagic shock model	Sprague-Dawley rats	males	↑ early survival, lung, liver and brain function	[128]
14.	Kidney ischemic/reperfusion injury model	Wistar rats	males	↑ blood IL-10 and TGF-β mRNA levels; ↑ direct correlation between IL-1β and TNF-α mRNA expression and IL-10 with TGF-β levels indicate the anti-inflammatory effect; ↓ kidney ischemic changes; ↓ serum creatinine level	[130]
15.	ARDS model	C57BL6 mice	males	↓ neutrophil influx into the lungs; ↓ local tissue destruction; ↓ the pulmonary and systemic inflammatory response	[131]
16.	Lung fibrosis model	C57BL/6J mice	males	↓ TGF-β1 in alveolar epithelial cells; alleviate lung fibrosis	[132]
17.	Gl. thymus model	Wistar rats	males and female	↑ *SLC5A8* gene expression in gonad intact female thymocytes; ↓ *SLC5A8* gene expression in gonad intact male thymocytes	[133]
	**Cell model *in vitro***				
18.	BHK-21 cells	Baby hamster kidney fibroblasts	unknown	↓ replication of enveloped viruses	[134]
19.	Vero cells	African green monkey kidney	unknown	↓ replication of enveloped viruses	[135]
20.	RAW264.7 macrophage cells	Mice	unknown	↓ macrophage-mediated Th1 effector, ↑ Th2 effector cell activation, affects the phenotype and function of macrophage (M1/M2)	[136]
21.	Bone marrow-derived macrophages	BALB/c mice	females	↓ the production of TNF-*α*, IL-6	[137]
22.	Bone marrow–derived primary macrophages (BMMs)	C57BL/6 BALB/c mice	females	polarises macrophages from a pro-inflammatory M1 to an anti-inflammatory M2 phenotype; ↓ IL-12 production and TNF-α by LPS-induced macrophage, ↑ IL-10 expression	[121]
23.	Alveolar epithelial cell line	A549	unknown	↓ TGF-β1-induced EMT in alveolar epithelial cells	[132]
24.	PBMCs of healthy subjects	Human	unknown	↑ apoptosis of normal human CD4^+^ and CD8^+^ T cells	[138]
25.	Monocytic leukemia THP-1 cells	Human	unknown	↓ LPS-induced production of TNF-α, IL-6	[126]
26.	Dendritic cells derived from monocytes of healthy blood donors’	Human	unknown	↓ monocyte differentiation into DCs; ↓ secretion of IL-8, IL-1b, IL-6, TNF-α and IL-10	[134]
27.	Healthy blood donors T CD8+ lymphocytes	Human	males + females (combined)	↓ in cellular proliferation	[139]

↓ decreased; ↑ increased.

**Table 2 biomedicines-10-00962-t002:** VPA treatment effect on thrombogenesis.

#	Thrombogenesis Related Factor	Cells/Animals/Human	Sex	VPA Treatment Effect	Ref.
1.	Complement C3	HepG2 cells	unknown	↓ *C3* gene expression	[209]
2.	t-PA	Human umbilical vein endothelial cells	unknown	↑ t-PA production	[210]
3.	ICAM-1 expression	Human umbilical vein ECs and human coronary artery EC	unknown	↓ ICAM-1 expression	[83]
4.	Platelets number	C57BL/6 mice	unknown	↓ platelets count	[200]
5.	Vascular t-PA	C57BL/6 mice	males	↑ endothelial vascular t-PA production; ↓ fibrin accumulation in response to vascular injury	[201]
6.	E-selectin and ICAM-1	Sprague–Dawley rats with subarachnoid hemorrhage induced vasospasm	males	↓ the E-selectin and ICAM-1 level	[211]
7.	Platelets number	Epileptic adult patients and healthy control	men and women	relationship between rising plasma VPA level and reduced platelet counts, with female sex additional risk factor	[203]
8.	Arachidonate cascade thromboxane A2 in platelets	Epileptic adult patients and healthy control	men	↓ activity of the arachidonate cascade in platelets; ↓ the cyclooxygenase pathway; ↓ synthesis thromboxane A2	[199]
9.	Von Willebrand factor:antigen	Epileptic children patients and healthy control	male + female (combined)	↓ concentration in blood serum	[205,206]
10.	Protein C	Epileptic children patients and healthy control	male + female (combined)	↓ concentration in blood serum	[205,207]
11.	Protein S	Epileptic children patients and healthy control	male + female (combined)	↓ concentration in blood serum	[207,208]
12.	Antithrombin III	Epileptic children patients and healthy control	male + female (combined)	↓ concentration in blood serum	[205,206]
13.	Prothrombin time	Epileptic children patients and healthy control	male + female (combined)	↓ concentration in blood serum	[205,206]
14.	Activated partial thromboplastin time	Epileptic children patients and healthy control	male + female (combined)	↓ concentration in blood serum	[205,206,207]

↓ decreased; ↑ increased.

## Data Availability

No new data were created or analyzed in this study. Data sharing is not applicable to this article.

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
