# Peer review of "SARS-CoV-2 Infection, Sex-Related Differences, and a Possible Personalized Treatment Approach with Valproic Acid: A Review"

_biomedicines, 2022, doi:10.3390/biomedicines10050962_

Round 1

Reviewer 1 Report

Authors present an interesting and quite expansive overview on the mechanism of action of valproic acid in relation to SARS-CoV-2 infection. The review is well written however there are some minor problems:

  1. Sex is biological, and gender is social. In context of the study, I believe the word sex rather than gender should be used.
  2. Abstract (line 28): "pharmacological mechanisms suggesting that VPA is promising for treating COVID-19". Either be more specific or omit this phrase since the following sentence specifies why VPA has a potential 
  3. line 167: "infection level" = perhaps "load"
  4. line 279: "settling" = perhaps "limiting"

Author Response

Answers to comments of Reviewer

Comments and Suggestions for Authors

The authors present an interesting and quite expansive overview of the mechanism of action of valproic acid in relation to SARS-CoV-2 infection. The review is well written however there are some minor problems:

Sex is biological, and gender is social. In the context of the study, I believe the word sex rather than gender should be used.

Answer

Corrected by the note.

Abstract (line 28): "pharmacological mechanisms suggesting that VPA is promising for treating COVID-19". Either be more specific or omit this phrase since the following sentence specifies why VPA has a potential.

Abstract (line 28): 'pharmacological mechanisms suggesting that VPA should be studied for its efficacy in COVID-19'.

Answer

Thank you for your comment. We have changed the sentence to reflect it.

line 167: "infection level" = perhaps "load"

Answer

Corrected by the note.

line 279: "settling" = perhaps "limiting"

thus settling mitochondrial energy production

Answer

We left out the word "settling" because our study shows that SLC5A8 expression is downregulated in male rats' thymocytes and significantly upregulated in thymocytes of gonad-intact female rats. The mechanisms behind this are unknown and deserve further investigation. We, therefore, think that the word "settling" is more appropriate in this situation.

The authors are sincerely grateful to the Reviewer for his positive evaluation of the manuscript and for his comments and suggestions.

Reviewer 2 Report

Comments for authors:

Title:

SARS-CoV-2 infection, sex-related differences, and a possible personalized treatment approach with valproic acid. A Review

Authors:

Stakisaitis D., Kapocius L., Valanciute A., Balnyte I., Tamosuitis T., Vaitkevicius A., Suziedelis K., Urboniene D., Tatarunas V., Kilimaite E., Gecys D., Lesauskaite V.

Manuscript ID: biomedicines-1673453

Objective:

The review summarizes the current knowledge of valproic acid as a promising therapy for COVID-19 with a special focus on sexual differences, bioavailability, anti-thrombotic, immunomodulatory, anti-inflammatory mechanisms, both against SARS-CoV-2 entry as well as COVID-19 associated complications like inflammation, thrombosis and endothelial dysfunction.

Comments:

Page 2 – line 71 “[35, 35]” and page 4 – line 152 “[68, 87, 87]”

References should be cited only once, i.e., [35] and [68, 87].

Chapter titles 3 and 6 should be written in boldface type.

Page 3 – line 145:

“… transmembrane serine protease 2 (MPRSS2) receptor …” - Shouldn't the abbreviation be (TMPRESS2)?

Page 4 – line 163:

… binding to AEC2… should read …binding to ACE2…

Page 4 – line 188:

… of HDAC2 RNR … should probably read … HDAC2 RNA…

References:

The references section needs to be revised thoroughly.

In general, it should be noted that some preprints (without peer review) are cited, which raises the question of whether this is indispensable with a total number of 266(!) references.

Some titles are written in italic or capital letters – all citations should be written consistently.

Reference No. 14 – Journal name should read “Pathologe” - instead of Pathol.

Ref 79 – “Mol. Ther. Methods Clin. Dev.” - instead of - Mol. Ther. – Methods Clin. Dev.

Ref 136 – Journal name should read “Diseases” - instead of Dis. Basel Switz.

Ref 102 & 231 – Journal name should read “Hypertension” - instead of Hypertens

Detailed Volume and/or page numbers are missing in Ref. 19, 50, 77, 83, 84, 145, 

205, 212, 243, 245.

Journal designations are inadequate at Ref. 29, 33, 34, 40, 63, 102, 106, 121, 125, 127, 138, 182, 186, 198, 204, 209, 213, 218, 231, 232, 239.

The number of co-authors should always be complete and consistent according to the journal's guidelines. The following references must be improved in this respect – Ref. 50, 58, 62, 77.

Some references are incomplete and need to be completed: Ref. 22, 31, 32, 52 and 265.

In Ref. 70 all authors are indicated by a single Christian name – this should be amended.

Ref. 60: tRNA – instead of TRNA

Ref. 62: Krämer – instead of Kamer

Ref. 87: Pöhlmann instead of Pohlmann

Ref. 142: First author is Harikrishnan K.N. – instead of K.N., H.

Ref 77 and 85: First names have been swapped with surnames - this needs to be corrected, i.e., Hu Y., Li X. – instead of Ying H., Xueyan L. (77); as well as Singh S. instead of Shweta S. (85)

Ref. 78: citation is incomplete – it is published at bioRxiv (preprint) – this needs to be adjusted.

Ref. 145: should read 2022, 36, 20587384211051954 – instead of 205873842110519

Ref. 162: should read doi:10.1001/jamainternmed.2019.7194 – instead of “jamaintermmed”

Ref. 178: was published in Immunology 10.1111/imm.13223; or as a preprint in medRxiv doi:10.1101/2020.03.12.20034736

Ref. 251: “VALPROIC ACID IMPROVES BETA CELL FUNCTION” should be removed.

Author Response

Comments and Suggestions for Authors

Comments for authors:

 Title:

SARS-CoV-2 infection, sex-related differences, and a possible personalized treatment approach with valproic acid. A Review

 Authors:

Stakisaitis D., Kapocius L., Valanciute A., Balnyte I., Tamosuitis T., Vaitkevicius A., Suziedelis K., Urboniene D., Tatarunas V., Kilimaite E., Gecys D., Lesauskaite V.

Manuscript ID: biomedicines-1673453

Objective:

The review summarizes the current knowledge of valproic acid as a promising therapy for COVID-19 with a special focus on sexual differences, bioavailability, anti-thrombotic, immunomodulatory, anti-inflammatory mechanisms, both against SARS-CoV-2 entry as well as COVID-19 associated complications like inflammation, thrombosis and endothelial dysfunction 

First of all, we would like to thank the Reviewer very sincerely for his great work and for being so thorough. It was a great lesson for us. Thank you very much!

 Comments:

 Page 2 – line 71 “[35, 35]” and page 4 – line 152 “[68, 87, 87]”

References should be cited only once, i.e., [35] and [68, 87].

Answer

Thank you for the comment. Correction - duplicates were deleted.

Chapter titles 3 and 6 should be written in boldface type.

Answer

Corrected by note.

Page 3 – line 145:

“… transmembrane serine protease 2 (MPRSS2) receptor …” - Shouldn't the abbreviation be (TMPRESS2)?

 Answer

Corrected by note

 Page 4 – line 163:

… binding to AEC2… should read …binding to ACE2…

 Answer

Corrected by note

Page 4 – line 188:

… of HDAC2 RNR … should probably read … HDAC2 RNA…

Answer

Corrected by note.

 References:

The references section needs to be revised thoroughly.

In general, it should be noted that some preprints (without peer review) are cited, which raises the question of whether this is indispensable with a total number of 266(!) references.

 Answer

Thanks very much for the important comment. The References section has been revised in line with the comment. All corrections have been made with Track Changer and a comment has been added to each reference.

 Preprints (former sources 83 and 84) have been removed and replaced by an article by the same authors - reference 83 in the revised manuscript.

Some titles are written in italic or capital letters – all citations should be written consistently.

Answer

Corrected by note.

Reference No. 14 – Journal name should read “Pathologe” - instead of Pathol.

Answer

Corrected by note.

Ref 79 – “Mol. Ther. Methods Clin. Dev.” - instead of - Mol. Ther. – Methods Clin. Dev.

Answer

Corrected by note.

Ref 136 – Journal name should read “Diseases” - instead of Dis. Basel Switz.

Answer

Corrected by note.

Ref 102 & 231 – Journal name should read “Hypertension” - instead of Hypertens

Answer

Corrected by note.

 Detailed Volume and/or page numbers are missing in Ref. 19, 50, 77, 83, 84, 145, 

205, 212, 243, 245.

Answer

50, 77, 205, 212 corrected in accordance with the note.

243 and 245 repeated the same source (245 removed).

83 and 84 are preprints - removed and replaced by an article by the same authors.

19, 145, 243 have not been corrected as they are cited by journal citation.

Journal designations are inadequate at Ref. 29, 33, 34, 40, 63, 102, 106, 121, 125, 127, 138, 182, 186, 198, 204, 209, 213, 218, 231, 232, 239.

Answer

Thank you for the comment. We have corrected all journal titles accordingly. 

The number of co-authors should always be complete and consistent according to the journal's guidelines. The following references must be improved in this respect – Ref. 50, 58, 62, 77.

Answer

Corrections made in accordance with the remark

Some references are incomplete and need to be completed: Ref. 22, 31, 32, 52 and 265.

Answer

Corrections made in accordance with the remark

In Ref. 70 all authors are indicated by a single Christian name – this should be amended.

Answer

Ref. 60: tRNA – instead of TRNA

Answer

Corrected according to note.

 Ref. 62: Krämer – instead of Kamer

Answer

Corrected by note.

Ref. 87: Pöhlmann instead of Pohlmann

Answer

Corrected by note.

Ref. 142: First author is Harikrishnan K.N. – instead of K.N., H.

Answer

Corrected by note.

 Ref 77 and 85: First names have been swapped with surnames - this needs to be corrected, i.e., Hu Y., Li X. – instead of Ying H., Xueyan L. (77); as well as Singh S. instead of Shweta S. (85)

Answer

Corrected by note.

Ref. 78: citation is incomplete – it is published at bioRxiv (preprint) – this needs to be adjusted.

Answer

Corrected by note.

 Ref. 145: should read 2022, 36, 20587384211051954 – instead of 205873842110519

 2022, 36, 205873842110519

Answer

Corrected by note.

Ref. 162: should read doi:10.1001/jamainternmed.2019.7194 – instead of “jamaintermmed”

Answer

Corrected by note.

Ref. 178: was published in Immunology 10.1111/imm.13223; or as a preprint in medRxiv doi:10.1101/2020.03.12.20034736

Answer

Was published in Immunology. Corrected by note.

Ref. 251: “VALPROIC ACID IMPROVES BETA CELL FUNCTION” should be removed.

Answer

Corrected by note.

Round 2

Reviewer 2 Report

Chapter titles 3 and 6 should be written in boldface type.

Page 3 – line 145:

“… transmembrane serine protease 2 (MPRSS2) receptor …” - Shouldn't the abbreviation be (TMPRESS2)?

Page 4 – line 163:

… binding to AEC2… should read …binding to ACE2…

Page 4 – line 188:

… of HDAC2 RNR … should probably read … HDAC2 RNA…

If the authors are adamant that their spelling is correct, they need to explain TMPRESS2, AEC2 and RNR at the first mention.

Ref.: 144 – Jukneviciene 2022: The identification number is 20587384211051954 – the authors mentioned 205873842110519 – this needs improvement.

Ref. 84: Singh S. and Singh, K.K. instead of Krishna K,S.

Author Response

Replies to Reviewer's comments

Thank you for your comments on the outstanding corrections to previous comments. We corrected the deficiencies then, but it turned out that we hadn't saved them and submitted the manuscript without checking. We therefore apologise. We have corrected all the deficiencies mentioned.

Chapter titles 3 and 6 should be written in boldface type.

Answer

Corrected by comment.

Page 3 – line 145:

“… transmembrane serine protease 2 (MPRSS2) receptor …” - Shouldn't the abbreviation be (TMPRESS2)?

Answer

Corrected by comment.

 Page 4 – line 163:

… binding to AEC2… should read …binding to ACE2…

Answer

Corrected by comment.

 Page 4 – line 188:

… of HDAC2 RNR … should probably read … HDAC2 RNA…

 If the authors are adamant that their spelling is correct, they need to explain TMPRESS2, AEC2 and RNR at the first mention.

Answer

Corrected by comments.

Ref.: 144 – Jukneviciene 2022: The identification number is 20587384211051954 – the authors mentioned 205873842110519 – this needs improvement.

Answer

Corrected by comment.

 Ref. 84: Singh S. and Singh, K.K. instead of Krishna K,S.

Answer

Corrected by comment.
